# Two-Level Planning of Customized Bus Routes Based on Uncertainty Theory

**Bing Zhang** [1,*]**, Zhishan Zhong** [1]**, Zi Sang** [1]**, Mingyang Zhang** [2] **and Yunqiang Xue** [1]

1  School of Transportation and Logistics, East China Jiaotong University, Nanchang 330013, China; zhongzhishan997@163.com (Z.Z.); sz1783195427@163.com (Z.S.); xueyunqiang@ecjtu.edu.cn (Y.X.)
2  School of Architecture and Civil Engineering, Jiangxi V&T College Communications, Nanchang 330013, China; mingyang0805@126.com
*  Correspondence: zhangbing@ecjtu.edu.cn; Tel.: +86-13870671586

**Abstract:** The optimization problem of customized bus routes is affected by uncertain factors in reality; therefore, this paper introduces uncertainty theory to study the above problem. A two-level planning model that takes the maximum total revenue of the bus company as the upper-level goal and the minimum total travel cost of passengers as the lower-level goal is established, using uncertainty theory to study and solve practical problems with uncertain factors. The genetic algorithm is used to solve the model, and the feasibility of the model is verified through a case study. The research results show that the application of the two-level model of customized bus route planning based on uncertain vehicle operating time established in this paper to customize bus route planning can take into account the travel needs of passengers and high-quality experiences while also bringing benefits to enterprises and achieving a win–win situation. The research in this article provides theoretical support for the optimization of customized bus routes.

**Keywords:** customized bus; uncertainty theory; route optimization; genetic algorithm; two-level planning





## 1. Introduction

Alleviating urban traffic problems by vigorously developing public transportation has become an effective measure widely adopted at home and abroad, and giving priority to public transportation has gradually become a sustainable transportation development strategy for large and medium cities in China. As a kind of personalized public transportation that has emerged in recent years, customized buses can make up for the shortcomings of conventional public transportation. Customized public transportation is a public transportation service mode between conventional public transportation and taxis. In the form of multiple people sharing transportation, its goal is to tailor public transportation services for people in similar areas with similar travel times and with similar travel needs. Its peculiar nature of fixed time, fixed point, and fixed person can meet the personalized travel needs of passengers.

Research on customized bus route planning at home and abroad mainly has the following goals: to reduce the time of customized bus operation, to save on customized bus operating costs, to provide passengers with better services, and to optimize comprehensive goals. Cederand Wilson [1] comprehensively considered the needs of traveling passengers and the profitability of operating companies when designing customized bus routes and proposed a data demand analysis method. KCS and MSP [2] regard the minimum total system cost as the optimization objective of the model, and the scale and service scope of the customized bus as decision variables; established a customized bus route optimization model; and found that the scale of customized bus needs to be larger than that of conventional buses with the same number of passengers. Nair and Miller-Hooks [3] designed a two-level planning model with customized bus lines on the upper layer and conventional bus lines on the lower layer. Tom V M and Mohan S [4] established a model that

considers both the total travel time of passengers and the total operating expenses of the bus company and uses genetic algorithms to find the best route and departure frequency. Pang Mingbao [5] and others built a customized bus two-level planning model based on game theory. The upper-level goal was the lowest operating cost of the bus company, and the lower-level goal was the highest travel efficiency for passengers. The model was solved by a meta-heuristic algorithm, and the best solution was designed. Guo Rongge [6] constructed a mixed integer programming model describing the problem to achieve dual decision-making optimization of routes and paths. The model aimed to maximize the total operating revenue and considered the characteristics of customized electric buses in the constraints. Second, in order to solve the model, a new adaptive large neighborhood search algorithm was designed, the corresponding initial solution generation rules and neighborhood search operators were proposed, and the effectiveness of the algorithm was verified through calculation examples. Gao Zhigang [7] established a customized bus route optimization model for the temporal and spatial distribution of passenger flow. The minimum travel cost of passengers was taken as the optimization objective, and the capacity and travel distance of the line were taken as constraints. The results showed that using this model to optimize the route of a customized bus can optimize the travel distance of the vehicle and can effectively reduce the travel cost of passenger flow. Hu Liege [8] and others established a two-level route planning model with the minimum number of vehicles as the upper-level target and the shortest operating mileage as the lower-level target for the customized bus operation mode of multi-point to multi-point operation. The passenger load factor and the detour distance of the vehicle were used as constraints. The model established on this basis is of great help to the planning of customized bus stations and the design of routes. Liu Wentian [9] and others established a customized bus route generation model that considers both customized bus operation companies and passengers. The Floyd shortest path algorithm (interpolation method) was used to generate the shortest path matrix based on the actual road network to find the shortest path. The results showed that the model can be applied to the actual urban public transportation system, providing new ideas for bus management. Ma Jihui [10] and others fully considered the travel needs of passengers and took customized buses under the "multi-start-multi-terminal" operation mode as the research object and conducted customized bus route planning research. They matched the number of passengers getting on and off the bus as the basis for planning the vehicle path. Wang Jian [11] and others took the minimum total operating mileage of all customized buses as the optimization goal; took the station constraints, vehicle capacity constraints, and travel time window constraints as constraints; and constructed a customized bus scheduling optimization model. The article analyzed the influence of the passenger's departure point and destination on the solution of the model and proposed to transform the above scheduling problem into a multi-traveling salesman problem using a virtual source station. Chen Mingming [12] considered the urban bus crew scheduling problem under the multi-parking factor and allowed the empty driving strategy of vehicles to increase the flexibility of the scheduling results. The flight time window was introduced in the article, and the basic interests of the crew were fully considered. Taking the cabin entry and exit costs, staying and waiting costs, and the minimization of empty driving costs as the objective function, a multi-park bus crew scheduling optimization model with time windows and a taboo search algorithm were established. Xu Wenpin [13] assumed that the demand of bus passengers obeys a certain normal distribution and used Monte Carlo method to simulate the bus demand matrix stochastically. Considering the perspectives of passengers, bus companies, and society, the total travel time of passengers, the vacant time of bus seats, the time difference between bus travel and private car travel, and bus fuel consumption were taken as the four objective functions of the model. Taking the weighted sum of tripartite benefits as the optimization goal of the network, a multi-objective optimization model of the bus network with uncertain passenger demand was constructed. Zhan Bin [14] took a single bus line as the research object, took into account the uncertainty of passenger demand, and established a two-layer model for determining the departure fre-



quency and for optimizing the non-uniform departure interval and systematically analyzed the bus dispatching problem.

In the research on customized bus routes in related literature, there are many optimization goals based on passenger travel time, passenger waiting time, vehicle operating mileage, corporate profitability, and corporate operating expenses. If only passengers are considered when optimizing the design of customized bus routes, often the result of a large number of vehicles, the comfort of passengers is guaranteed but bus companies will have invested too much cost. However, only considering the desires of the bus company will also result in a design with the route passing by many stops and with a long travel time, which does not to provide a good travel experience for passengers and makes the customized bus less attractive to travelers. Therefore, on the basis of previous studies, this paper comprehensively considers both the bus company and passengers and establishes a two-level planning model with the maximum total revenue of the bus company as the upper-level goal and with the minimum total travel cost of passengers as the lower-level goal. This model plans and designs customized bus routes.

It can be seen from reading the existing research that most research sets the running speed of the customized bus as a fixed value. Before the model is established, it is assumed that the vehicle will arrive at the stop at the rated speed, but this is not consistent with the actual situation. When the route designed under this assumption is put into practice, the passenger's satisfaction often decreases due to the low punctuality of the customized bus. In order to improve the punctuality of the customized bus and to make the model represent reality, this article introduces uncertainty theory to study the optimization of customized bus routes and to analyze the travel time of passengers between two stops as an uncertain variable. This method makes up for this shortcoming in existing research.

This paper therefore focuses on the variation in actual conditions due to factors such as the speed of the vehicle, road conditions during operation, the operating time of the vehicle at two stops, and the number of passengers getting on and off at each stop as uncertain variables and establishes a two-level planning model with the maximum total revenue for the bus company as the upper-level objective and the minimum total passenger travel cost as the lower-level objective. Theoretical support is provided for the optimization of customized bus routes.

## 2. Uncertain Theoretical Basis

Uncertainty theory was proposed and established by Liu Baoding, and many researchers subsequently studied it. At present, uncertainty theory has become a branch of axiomatic mathematics that is used to model belief degrees. Uncertainty theory is essentially a measurement theory, with four axioms of normativity, duality, subadditivity, and product measurement axioms [15]. Suppose that $\Gamma$ is a non-empty set, $L$ is the algebra of $\Gamma$, each element $\Lambda \in L$ is called an event, and an aggregate function $M : L \to [0,1]$ is called an uncertainty measure; its nature is as follows:

Property 1: For the uncertainty measure, if any event $\Lambda_1 \subset \Lambda_2$ is taken, there is $M\{\Lambda_1\} \leq M\{\Lambda_2\}$, then $M$ is called a monotonically increasing function;

Property 2: For the uncertainty measure $M$, the uncertainty measure of the empty set $\varnothing$ is zero, that is $M\{\varnothing\} = 0$; and

Property 3: if $M$ is an uncertainty measure, then for any event $\Lambda$, there is $0 \leq M\{\Lambda\} \leq 1$.

In addition, the uncertainty distribution and the inverse uncertainty distribution are defined:

**Definition 1.** *For an uncertain variable $\xi$, its uncertainty distribution $\Phi$ is defined as $\Phi(x) = M\{\xi \leq x\}$, where $x$ is any real number.*

**Definition 2.** *For an uncertain variable $\xi$, if its uncertainty distribution $\Phi$ has an inverse function $\Phi^{-1}(\alpha)$ and it exists and is unique to any $\alpha$, then $\Phi^{-1}$ is called the inverse uncertainty distribution of $\xi$.*

### 3. Customized Bus Route Optimization Model

*3.1. Problem Description*

As a new bus operating mode, customized buses can design routes in a targeted manner according to the departure station, arrival station, and expected travel time of passengers. Set the travel time of the customized bus between the two stations as the ratio of the distance to the speed of the two stations. In actual operation, the running speed of the vehicle between the two stations is not constant. Therefore, the run time between the two stations is an uncertain variable. People tend to ignore this situation when optimizing the route. The actual arrival time at each station thus does not match the prescribed time, and the punctuality rate of customized buses drops, causing dissatisfaction among passengers. In order to avoid this situation, this paper treats the run time of the vehicle between the two stations as an uncertain variable for analysis. In this way, the designed route operation can be more realistic, and passengers' satisfaction with the customized bus can be improved.

First, the modeling scene is set up: there are $K$ customized buses in a certain area, each bus can take $Q$ passengers, and there are $N$ customized bus stations in the area. Each customized bus has different routes and different stops through different routes. Assuming that the Nth stop passed by the Kth bus is $x_{kn}$, the travel path of the Kth bus is $x_{k_1} \rightarrow x_{k_2} \rightarrow \cdots \rightarrow x_{k_n}$, and the bus first goes to the various pick-up stations and then to the various drop-off stations.

Due to the randomness of the actual traffic state, it is difficult for the bus to arrive at the stop accurately according to the passenger's time requirement. Therefore, the passenger allows the bus pick-up time to have a certain time range fluctuation, which is called the time window in this article. Supposing the actual time when the Kth bus arrives at the station is $f_{k_n}$, the time window for passengers to allow the bus to arrive at the stop is $\left[ a_{x_{k_n}}, b_{x_{k_n}} \right]$; for example, the expected departure time of passengers is 7:30 a.m. and buses are allowed to arrive early or late by 5 min. The time window is [7:25,7:35], and the actual arrival time of the bus may be 7:32.

*3.2. Construction of the Model*

3.2.1. Model Construction

1. The travel background of the model is the morning peak commuting time period. In actual application, if it is the evening peak commuting time period, the model can be used by slightly modifying the model;
2. The expected travel time and expected arrival time of passengers at various stops passing by the customized bus are known;
3. The customized bus route is to go to each pick-up station in the route first and then to each drop-off station, the bus only stops at the pick-up and the drop-off station and does not stop midway.
4. Each station is served by only one customized bus, and the customized bus model and the approved passenger capacity are known; and
5. No consideration is given to the pick-up and drop-off time of passengers at various stations.

3.2.2. Upper Target

The upper-level model starts by maximizing the interests of the bus company, by increasing the fare revenue of the bus company, by reducing the cost of customized bus operation, and by maximizing the revenue of the enterprise. In order to achieve this goal, this article analyzes the bus company's profit maximization from the three aspects of the bus company's fare revenue, fixed costs, and operating costs.

Fare revenue: The revenue of the bus company mainly comes from the ticket sales of customized buses, and the fare of customized buses is set as $\rho$, the number of passengers

boarding at stop $x_{k_n}$ is $h_{x_{k_n}}$, and the revenue of the bus company is the sum of the fares of all passengers:

$$P_1 = \rho \sum_{k=1}^{K} \sum_{n=1}^{N} hx_{k_n} \tag{1}$$

Fixed cost: The fixed cost of a customized bus consists of the fee for the bus, the parking fee of the parking lot, and other costs. Supposing a fixed cost for each customized bus $C_1$, a total of K customized buses are set up in the area and the total fixed cost of the bus company is as follows:

$$P_2 = C_1 K \tag{2}$$

Operating costs: Customized buses also incur certain costs during operation, such as vehicle fuel costs, repair and maintenance costs, and depreciation costs, etc. These costs are collectively referred to as operating costs. Set the operating cost of bus per kilometer as $C_2$ and the total mileage of all customized buses as $L$, and the total operating cost of the bus company is as follow:

$$P_3 = C_2 L \tag{3}$$

Set the mileage between station $x_{kn}$ and station $x_{kn+1}$ as $l_{x_{k_n} x_{k_{n+1}}}$, then the total mileage of the Kth bus is as follows:

$$l_k = \sum_{n=1}^{N} l_{x_{k_n} x_{k_{n+1}}} \tag{4}$$

In summary, the objective function of the upper model is as follows:

$$\max P_1 = P_1 - P_2 - P_3 = \rho \sum_{k=1}^{K} \sum_{n=1}^{N} h_{x_{k_n}} - C_1 K - C_2 L \tag{5}$$

### 3.2.3. Lower Target

The lower model starts by maximizing the interests in passengers, by reducing the travel and waiting time of passengers, and by improving the comfort of buses, thereby ensuring that more passengers choose customized bus travel. The travel choice of passengers mainly depends on the time and money spent on the trip. The travel time also includes the cost of waiting time and travel time. This article analyzes the maximization of passenger benefits from three aspects: waiting time cost, travel time cost, and fare cost.

Waiting time cost: Since each station has a time window for the arrival of customized buses, it is necessary to discuss the time for each customized bus to arrive at different stations. Supposing that the departure time of the Kth bus from the first stop is $t_{x_{k_1}}$, the travel time $t_{x_{k_1} x_{k_2}}$ of the first stop to the second stop is an uncertain variable and the time for the Kth bus to arrive at stop $x_{k_2}$ is as follows:

$$f_{x_{k_2}} = t_{x_{k_1}} + t_{x_{k_1} x_{k_2}} \tag{6}$$

In actual operation, some customized buses may arrive at the station outside the time window, so this article stipulates the following:

1.  The customized bus arrives at the station earlier than the time specified in the time window and must wait until the earliest time allowed by the passenger to depart from the station;
2.  The customized bus arrives at the station within the time specified in the time window and can directly pick up and drop off passengers and depart from the station;
3.  If the customized bus arrives at the station later than the time specified in the time window, it can directly pick up passengers and depart from the station.

The time of Kth bus to arrive at station $x_{k_{n+1}}$ is as follows:

$$f_{x_{k_{n+1}}} = \max\left\{ f_{x_{k_n}}, a_{x_{k_n}} \right\} + t_{x_{k_n} x_{k_{n+1}}} \tag{7}$$

Therefore, for the Kth bus, since its run time $t_{x_{k_n} x_{k_{n+1}}}$ between station $x_{k_n}$ and station $x_{k_{n+1}}$ is an uncertain variable, the time $f_{k_n}$ when it arrives at station $x_{k_n}$ is also an uncertain variable and the corresponding inverse uncertainty distribution is as follows:

$$\Psi^{-1}_{x_{k_n}} = \max\left\{\Psi^{-1}_{x_{k_{n-1}}}, a_{x_{k_n}}\right\} + \Phi^{-1}_{x_{k_n} x_{k_{n+1}}} \tag{8}$$

The time for each customized bus to reach different stops can be calculated according to the above formula.

Every passenger has an expected arrival time of the vehicle when traveling. Due to the uncertainty of the run time of the bus between the two stations, the actual arrival time of the bus deviates from the expected arrival time of the passenger. The difference between the passenger's expected arrival time and the actual arrival time of the vehicle is the passenger's wait time at the station. It is stipulated that, if the bus arrives at the station before the passenger's expected arrival time, the wait time is 0. If the bus arrives at the stop after the passenger's expected arrival time, the wait time is the difference between the two. In order to allow passengers to have a better travel experience, this paper sets the minimum wait time cost of all passengers as one of the goals of the lower-level model, which is expressed by the following formula:

$$P_4 = \sum_{k=1}^{K} \sum_{n=1}^{N} \max\left\{t_{x_{k_n}} - \frac{a_{x_{k_n}} + b_{x_{k_n}}}{2}, 0\right\} \tag{9}$$

Travel time cost: In terms of travel speed, the speed of customized bus travel is slower than car travel and the travel time is also longer. In order to quantify how much more travel time is spent by passengers who choose customized bus travel than those who choose car travel, this article sets a total of $M$ passengers on the customized bus, the starting site of guest $m$ as $x_{k_m}$, the destination site as $x'_{k_m}$, the distance between the two stations as $l_{x_{k_m} x'_{k_m}}$, and the travel time of the passenger on the customized bus as $t_{x_{k_m} x'_{k_m}}$. Supposing the average speed of car travel is $v$, the time when the passenger chooses the car to travel is $\frac{l_{x_{k_m} x'_{k_m}}}{v}$ and the difference between the two travel times is the travel time cost of the passenger:

$$P_5 = \sum_{m=1}^{M} \left(t_{x_{k_m} x'_{k_m}} - \frac{l_{x_{k_m} x'_{k_m}}}{v}\right) \tag{10}$$

Fare cost: Set the fare of the customized bus as $\rho$ and the number of passengers on the bus at stop $x_{k_n}$ as $h_{x_{k_n}}$; then, the sum of the fare cost of all passengers is as follows:

$$P_6 = \rho \sum_{k=1}^{K} \sum_{n=1}^{N} h_{x_{k_n}} \tag{11}$$

In summary, the objective function of the lower model is as follows:

$$\min P_2 = P_4 + P_5 + P_6 = \sum_{k=1}^{K} \sum_{n=1}^{N} \sum_{m=1}^{M} \left[\max\left\{f_{x_{k_n}} - \frac{a_{x_{k_n}} + b_{x_{k_n}}}{2}\right\} + \left(t_{x_{k_m} x'_{k_m}} - \frac{l_{x_{k_m} x'_{k_m}}}{v}\right) + \rho h_{x_{k_n}}\right] \tag{12}$$

### 3.2.4. Constraints

1. Upper objective function constraints

Investment constraints: In order to avoid solving the optimal solution to the model, the investment value of the bus company is increased indefinitely, which does not match the actual situation, and it is necessary to restrict the investment budget of the bus company, which is as follows:

$$P_2 + P_3 < \varsigma_{max} \tag{13}$$

Revenue constraint: In order to ensure the revenue of the bus company and to prevent the final solution from making the expenditure of the bus company greater than the revenue, it is necessary to constrain the bus company's revenue to be positive, which is as follows:

$$P_1 > 0 \tag{14}$$

Route length constraints: The line length of customized buses should also be subject to certain restrictions. For society, a too short customized bus line increases the number of customized buses and increases road congestion. For bus companies, a short line length will increase its investment value but reduce its benefit value. Therefore, it is necessary to constrain the route length of the customized bus to meet the shortest operating length of the route, which is as follows:

$$l_k > l_{min} \tag{15}$$

Vehicle capacity constraints: In order to ensure the comfort of passengers on the bus, a customized bus is set up for one person. Due to the uncertainty of passengers getting on and off at each station, this article restricts the total number of passengers on the bus. Supposing the number of passengers boarding the Kth bus at station $x_{k_1}$ is $h_{x_{k_1}}$, the number of passengers on the bus when departing from station $x_{k_1}$ is $q_{x_{k_1}}$, which is as follows:

$$q_{x_{k_1}} = h_{x_{k_1}} \tag{16}$$

Similarly, the number of passengers on the bus when departing from the station $x_{k_n} x_{k_n}$ is as follows:

$$q_{x_{k_n}} = q_{x_{k_{n-1}}} + h_{x_{k_n}} \tag{17}$$

The number of passengers on each customized bus can be calculated according to the above formula.

2. Lower objective function constraints

Passenger ride constraint: In order to avoid the phenomenon in which the passenger's pick-up point and the drop-off point are served by different customized buses in the solution result, a constraint is imposed on the passenger's ride, stipulating that, if the customized bus passes the passenger's pick-up point, it must pass the passenger's drop-off point. Set decision variables $y(x_{k_n})$;

$$y(x_{k_n}) = \begin{cases} 1, \text{ customized bus passes through stop } x_{k_n} \\ 0, \text{ customized bus does not pass through stop } x_{k_n} \end{cases} \tag{18}$$

The constraints are as follows:

$$y(x_{k_n}) = y\left(x'_{k_m}\right) \tag{19}$$

Punctuality constraint: In order to ensure the punctuality of the customized bus operation, it is expected that the passenger at each stop will be served within the time window they determined with a confidence level of $\alpha$, and the customized bus arrives at station $x_{k_n}$ between time windows $\left[a_{x_{k_n}}, b_{x_{k_n}}\right]$ with a confidence level of $\alpha$, which can be expressed as follows:

$$M\left\{a_{x_{k_n}} \le T_{x_{k_n}} \le b_{x_{k_n}}\right\} \ge \alpha \tag{20}$$

Vehicle capacity constraint: In order to ensure the comfort of the customized bus, the number of passengers $q_{x_{k_n}}$ on board is expected to be less than the rated number of passengers $Q$ when the customized bus departs from station $x_{k_n}$. This can be expressed by the following formula:

$$q_{x_{k_n}} \le Q \tag{21}$$

### 3.2.5. Model Building

Synthesizing the above analysis, the two-level planning model for the design of customized bus routes based on the uncertainty of vehicle run time is described as follows:

Upper level model:

$$maxP_1 = \rho \sum_{k=1}^{K} \sum_{n=1}^{N} h_{x_{k_n}} - C_1 K - C_2 L s.t. \ p_1 + p_2 < \varsigma_{max} P_1 > 0 l_k > l_{min} q_{x_{k_n}} \leq Q \quad (22)$$

Lower level model:

$$minP_2 = \sum_{k=1}^{K} \sum_{n=1}^{N} \sum_{m=1}^{M} \left[ max\left\{ \frac{a_{x_{k_n}} + b_{x_{k_n}}}{2} - T_{x_{k_n}}, 0 \right\} + \left( t_{x_{k_m} x_{k_{m'}}} - \frac{l_{x_{k_m} x_{k_{m'}}}}{v} \right) + \rho h_{x_{k_n}} \right]$$
$$s.t. \ y(x_{k_m}) = y\left( x'_{k_m} \right)$$
$$M\left\{ a_{x_{k_n}} \leq T_{x_{k_n}} \leq b_{x_{k_n}} \right\} \geq \alpha q_{x_{k_n}} \leq Q \quad (23)$$

## 4. Solution of the Model

### 4.1. Determining the Uncertainty Distribution

Since the model contains uncertain variables, to solve the model, we must first judge the distribution status obeyed by the uncertain variables and then use uncertainty theory to derive its inverse uncertainty distribution. In order to simplify the calculation process, this paper assumes that the vehicle run time obeys the normal uncertainty distribution.

Normal uncertain variables: If an uncertain variable $\zeta$ obeys a normal uncertain distribution, then its uncertainty distribution is as follows:

$$\Phi(x) = \left( 1 + exp\left( \frac{\pi(e - x)}{\sqrt{3}\sigma} \right) \right)^{-1}, x \in R \quad (24)$$

where $e$ and $\sigma$ are constants and $\sigma$ is greater than 0. Denote the above distribution as $N(e, \sigma)$. Then, the inverse uncertainty distribution of the normal uncertainty distribution $N(e, \sigma)$ is as follows:

$$\Phi^{-1}(\alpha) = e + \frac{\sigma\sqrt{3}}{\pi} ln \frac{\alpha}{1 - \alpha} \quad (25)$$

The distribution of uncertain variables at this confidence level is obtained by giving the desired confidence level and by transforming the uncertainty distribution into an inverse uncertainty distribution.

### 4.2. Genetic Algorithm Design

Compared with other algorithms, the genetic algorithm can process multiple individuals in the group at the same time, can evaluate multiple solutions in the search space, can reduce the risk of falling into the local optimal solution, and can facilitate the global optimization, and the algorithm easily realizes parallelization. The genetic algorithm differs from other traditional search methods in the following respects: (1) they search among a population of points and not a single point, (2) they work not on the variables but on the code of the variable, (3) the transition scheme is probabilistic instead of deterministic, and (4) they use objective function information in place of gradient information.

The genetic algorithm is a relatively convenient operation with high solution accuracy, strong optimization-seeking ability, good constraint handling ability, and high method maturity and is more suitable for solving complex multi-objective optimization problems. Given that the model designed in this paper is a multi-objective optimization model, the genetic algorithm is selected to solve the model, and the specific steps and flow charts are as follows.

1. Encoding. In this paper, the coding method is set to natural number coding. In the chromosome encoded with natural number, each gene can represent a node, and at the same time, the order of different gene arrangement represents the sequence of nodes. The specific chromosome coding method is as follows: there are stations in the region, so that the natural numbers 1, 2, 3, . . . , represent stations, so that the chromosome composed of these natural numbers can be used to represent the custom bus travel routes.

2. Initial population. Generally, the initial population is determined by choosing a random selection method, and in this paper, the nearest neighbor heuristic method is used to determine the initial population in order to improve the efficiency of the algorithm. The specific steps of the method are as follows:

   STEP1: Randomly select a station from all stations as the starting point of the first bus;

   STEP2: Randomly selecting the next station from the remaining stations and determining whether the time window limit is satisfied;

   STEP3: If the time window limit is met, continue to determine if the passenger demand limit is met;

   STEP4: If the time window restriction is not met, the site will not be regarded as the next site so go back to step 2;

   STEP5: If the time window limit and passenger demands limit is met, the station is taken as the next station and steps 2–4 are repeated to continue the search for subsequent stations; and

   STEP6: If the time window limit is met, the passenger demand limit is not met, and additional stops would result in more passengers than the rated capacity of the customized bus, the stop will not be used as the next stop and will be transferred directly to the end point.

   After the route of one customized bus is generated, a random station from the remaining stations is used as the starting point of the second bus, and steps 2–6 are repeated to generate the routes of other customized buses. Follow the above steps repeatedly until the number of generated chromosomes meets the population size.

3. Fitness function. In order to evaluate whether the individuals in the population are good or not, the fitness function is established to evaluate the degree of goodness of the individuals. If the calculated individual fitness function value is higher, it indicates that each individual in the chromosome is better and that the overall structure of the chromosome is better and closer to the optimal solution. In order to express the degree of individual goodness more intuitively, this paper directly defines the fitness function of the chromosome as the objective function of the model.

4. Operator selection. The three operators of genetic algorithms are selection, crossover, and mutation, which simulate biological reproduction, mating, and genetic mutation, respectively.

   (1) For the selection operator, here, we use the roulette selection operator. The sum of the fitness values of all chromosomes is first calculated according to the rules of roulette selection.

$$fitness = \sum fitness\ i \tag{26}$$

Second, the cumulative fitness value of the chromosome is calculated.

$$pot\ i = \frac{fitness\ i}{fitness} \tag{27}$$

Calculate the cumulative fitness value of the chromosome again.

$$aof\ i = \sum_{j=1}^{j=i-1} pof\ j \tag{28}$$

Finally, select chromosomes based on random numbers.

(2) For the crossover operator, we use the arithmetic crossover operator. The definition takes one departure stop and its corresponding arrival stop as the basic unit of the chromosome, and there are K buses on the route, i.e., there are K initial populations. In real life, the number of stops on each customized bus route is generally small, so the enumeration algorithm can be used to crossover all the basic units of the K chromosomes to produce the next generation population. The principle that must be followed for crossover operations is that they are performed with basic units and are not allowed to crossover only one departure or arrival station. This article defines the following two types of crossover operations: (1) the self-crossing of two basic units of a single chromosome is the actual meaning of the adjustment of the start and end positions of a single customized bus line. (2) Each of the two chromosomes adopts a basic unit for crossover. The actual meaning is the exchange of a pair of starting points and ending points in two bus lines.

(3) Additionally, for the mutation, we use the uniform mutation operator. The uniform mutation operation involves replacing the original gene values at each locus in an individual coding string with a random number that fits within a range of uniform distribution with a small probability, respectively.

5.　Termination condition. Set the maximum number of evolutionary generations, and the calculation terminates when the genetic algorithm runs for the generation specified by this parameter.

The basic flow of the genetic algorithm is shown in Figure 1.

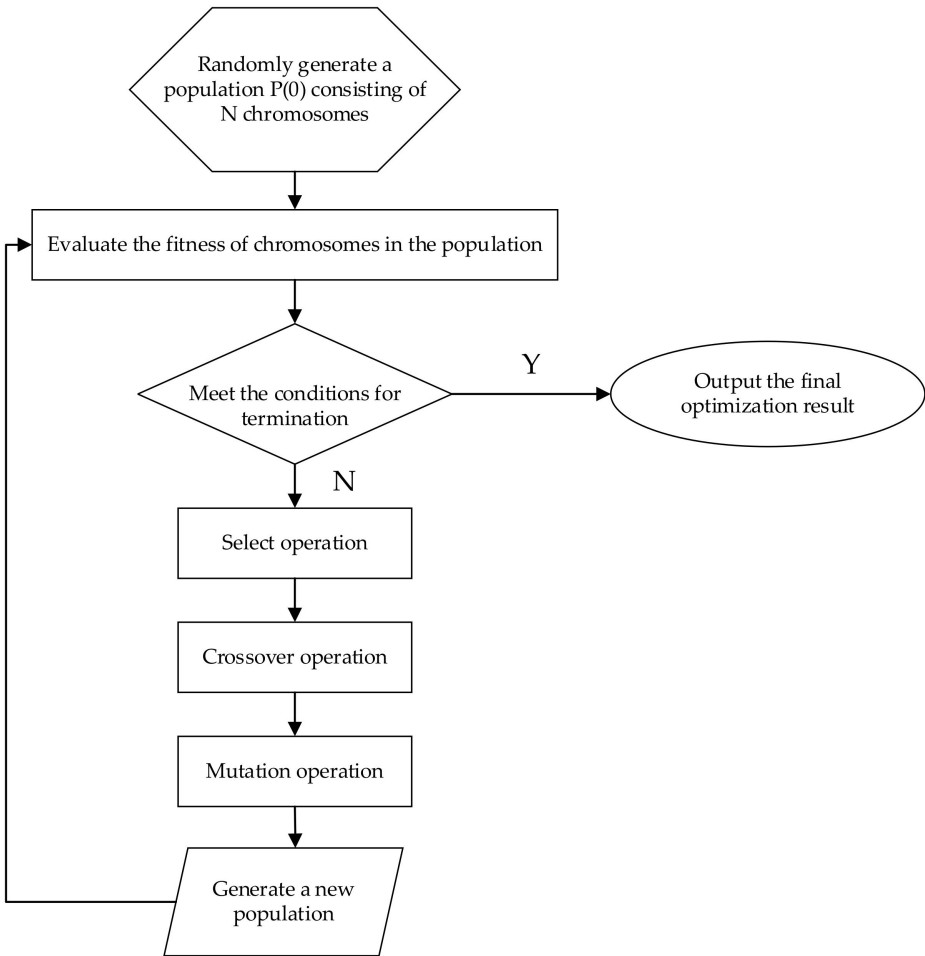

**Figure 1.** The basic process of genetic algorithm.

### 5. Case Study

In order to test the practicality of the model, the model established in this paper was applied to the development of custom bus routes in Nanchang City. Firstly, 120 passengers' travel-related information was collected, and the service range of the custom bus stops was set to 500 m. The 16 stops planned could serve 104 customers, of which 16 passengers exceeded the service range of the stops, and these passengers could be included in the recruitment of later routes. There are eight pick-up points and eight drop-off points in the area. According to the information collected from the passengers' departure, destination, expected ride time, expected arrival time, and other information to select custom bus stops, the number of people served and service time of each departure and destination stops are summarized in Table 1 below.

**Table 1.** Departure and destination station information.

| Bus Stop | Number of People | Time Window Limit |
|---|---|---|
| O1 | 17 | [6:50–7:00] |
| O2 | 13 | [7:15–7:25] |
| O3 | 12 | [7:00–7:10] |
| O4 | 16 | [7:10–7:20] |
| O5 | 11 | [7:25–7:35] |
| O6 | 14 | [7:05–7:15] |
| O7 | 10 | [7:10–7:20] |
| O8 | 11 | [7:20–7:30] |
| D1 | 14 | [7:45–7:55] |
| D2 | 7 | [7:55–8:05] |
| D3 | 9 | [8:05–8:15] |
| D4 | 17 | [7:55–8:05] |
| D5 | 22 | [8:05–8:15] |
| D6 | 11 | [7:55–8:05] |
| D7 | 13 | [8:05–8:15] |
| D8 | 11 | [8:15–8:25] |

In this case, the parameters involved in the model are calibrated in the context of the actual situation. In the model assumptions, the modeling scenario is based on morning peak commuting behavior, so the parameters are set as one-way values. In order to improve the comfort of passengers riding the customized bus, a customized bus with a rated capacity $Q$ of 40 passengers is used for the analysis. The fare of the customized bus is set at 5 CNY, which is the same as the fare of the existing customized bus. The fixed cost of bus operation is $C_1 = 100$ CNY/veh, the running cost is $C_2 = 1.66$ CNY/km. The distance between each station is shown in Table 2 below. The run time of the customized bus between the two stations obeys the normal uncertainty distribution $t_{x_{k_n} x_{k_{n+1}}} \sim N\left(2l_{x_{k_n} x_{k_{n+1}}}, 1\right)$, and the confidence level is taken as 0.9. Since the solution scenario is during the morning peak period, we referred to the real-time vehicle running speed on Baidu map and set the running speed of the small car as $v = 40$ km/h. For the constraints, the maximum investment value of the bus company was set to $\varsigma_{max} = 3000$ CNY, and the minimum value of the line length was $l_{min} = 5$ km. The population size was set to $R = 10$, the crossover probability was set to $P_C = 0.60$, the variation probability was set to $Pm = 0.03$, and the maximum number of evolutionary generations was set to $T = 60$.

**Table 2.** Distance between stops (km).

| Stop | O1 | O2 | O3 | O4 | O5 | O6 | O7 | O8 | D1 | D2 | D3 | D4 | D5 | D6 | D7 | D8 |
|------|-----|-----|-----|-----|-----|-----|-----|-----|-----|-----|-----|-----|-----|-----|-----|-----|
| O1 | 0 | 1.3 | 2.2 | 1.7 | 2.3 | 2.8 | 3.1 | 2.8 | 4.4 | 5.4 | 5.4 | 4.5 | 5.5 | 4.4 | 4.9 | 5.0 |
| O2 | 1.3 | 0 | 2.6 | 1.8 | 1.2 | 2.5 | 2.5 | 1.9 | 3.5 | 4.6 | 4.6 | 3.5 | 4.5 | 3.2 | 3.8 | 3.8 |
| O3 | 2.2 | 2.6 | 0 | 0.9 | 2.7 | 1.3 | 2.0 | 2.5 | 6.2 | 7.3 | 7.2 | 6.1 | 7.1 | 5.7 | 6.3 | 6.1 |
| O4 | 1.7 | 1.8 | 0.9 | 0 | 1.9 | 1.2 | 1.7 | 1.8 | 5.4 | 6.5 | 6.4 | 5.3 | 6.2 | 4.9 | 5.5 | 5.3 |
| O5 | 2.3 | 1.2 | 2.7 | 1.9 | 0 | 2.0 | 1.7 | 0.8 | 3.9 | 5.0 | 4.8 | 3.7 | 4.5 | 3.2 | 3.7 | 3.4 |
| O6 | 2.8 | 2.5 | 1.3 | 1.2 | 2.0 | 0 | 0.7 | 1.4 | 5.8 | 7.0 | 6.8 | 5.7 | 6.5 | 5.1 | 5.7 | 5.3 |
| O7 | 3.1 | 2.5 | 2.0 | 1.7 | 1.7 | 0.7 | 0 | 0.9 | 5.5 | 6.6 | 6.5 | 5.3 | 6.1 | 4.7 | 5.3 | 4.9 |
| O8 | 2.8 | 1.9 | 2.5 | 1.8 | 0.8 | 1.4 | 0.9 | 0 | 4.7 | 5.8 | 5.6 | 4.5 | 5.2 | 3.9 | 4.4 | 4.0 |
| D1 | 4.4 | 3.5 | 6.2 | 5.4 | 3.9 | 5.8 | 5.5 | 4.7 | 0 | 1.1 | 1.0 | 0.6 | 1.4 | 1.3 | 1.3 | 2.0 |
| D2 | 5.4 | 4.6 | 7.3 | 6.5 | 5.0 | 7.0 | 6.6 | 5.8 | 1.1 | 0 | 0.5 | 1.5 | 1.4 | 2.2 | 1.8 | 2.7 |
| D3 | 5.4 | 4.6 | 7.2 | 6.4 | 4.8 | 6.8 | 6.5 | 5.6 | 1.0 | 0.5 | 0 | 1.2 | 0.9 | 1.9 | 1.5 | 2.3 |
| D4 | 4.5 | 3.5 | 6.1 | 5.3 | 3.7 | 5.7 | 5.3 | 4.5 | 0.6 | 1.5 | 1.2 | 0 | 1.0 | 0.8 | 0.7 | 1.5 |
| D5 | 5.5 | 4.5 | 7.1 | 6.2 | 4.5 | 6.5 | 6.1 | 5.2 | 1.4 | 1.4 | 0.9 | 1.0 | 0 | 1.4 | 0.8 | 1.5 |
| D6 | 4.4 | 3.2 | 5.7 | 4.9 | 3.2 | 5.1 | 4.7 | 3.9 | 1.3 | 2.2 | 1.9 | 0.8 | 1.4 | 0 | 0.7 | 0.9 |
| D7 | 4.9 | 3.8 | 6.3 | 5.5 | 3.7 | 5.7 | 5.3 | 4.4 | 1.3 | 1.8 | 1.5 | 0.7 | 0.8 | 0.7 | 0 | 0.9 |
| D8 | 5.0 | 3.8 | 6.1 | 5.3 | 3.4 | 5.3 | 4.9 | 4.0 | 2.0 | 2.7 | 2.3 | 1.5 | 1.5 | 0.9 | 0.9 | 0 |

The set parameters and site information ware imported into MATLAB, and the genetic algorithm was applied to solve the final design. Line 1: The number of passengers served was 30, and there were 2 pick-up points, which were O1 and O2. There were three drop-off points, which were D1, D2, D3. Line 2: The number of passengers served was 39, and there were 3 pick-up points, which were O3, O4, and O5. There were two drop-off points, which were D4 and D5. Line 3: The number of passengers served was 35, and there were 3 pick-up points, which were O6, O7, and O8. There were 3 drop-off points, which were D6, D7, and D8.

The solutions using the conventional genetic algorithm and the improved genetic algorithm are listed in Table 3. Compared with traditional methods, the simulation method used in this paper reduces the total number of miles operated, thereby reducing the operating costs of the bus company and increasing revenue.

**Table 3.** Comparison of optimization schemes obtained by different algorithms.

| Comparison Index | Conventional Genetic Algorithm | Improved Genetic Algorithm |
|------------------|-------------------------------|---------------------------|
| Optimization | Customized bus 1: O1→O3→O8→D1→D4→D3 Customized bus 2: O6→O7 →D6→D7 Customized bus 3: O4→O2→O5→D2→D5→D8 | Customized bus 1: O1→O2→D1→D2→D3 Customized bus 2: O3→O4→O5→D4→D5 Customized bus 3: O6→O7→O8→D6→D7→D8 |
| Total Operating Mileage (km) | 28.2 | 21 |
| Operating Costs (CNY) | 46.812 | 34.86 |
| Fixed Costs (CNY) | 300 | 300 |
| Fare Revenue (CNY) | 520 | 520 |
| Occupancy Rate (%) | 100%, 60%, 100% | 75%, 97.5%, 87.5% |

Considering the actual road network characteristics, more lines are selected to set up bus lanes in order to make the fastest bus travel speed, so that passengers achieve the optimal travel experience. The final design of the line-specific direction schematic diagram is shown in Figure 2 below.

In this customized bus route design plan, three customized bus routes were planned to serve 104 passengers. When passenger information collection was conducted, information about 120 passengers was collected, and 16 passengers were beyond the scope of service in this plan. The total service rate of the bus routes reached 87%, and the coverage of the service was large, while the individual needs of passengers were fully considered. For each customized bus, the number of passengers served were 30, 39, and 35, respectively. Given the rated capacity of the customized bus is 40 passengers, the occupancy rate of each customized bus is 75%, 98%, and 88%, respectively, which is designed to ensure one per person without wasting resources.

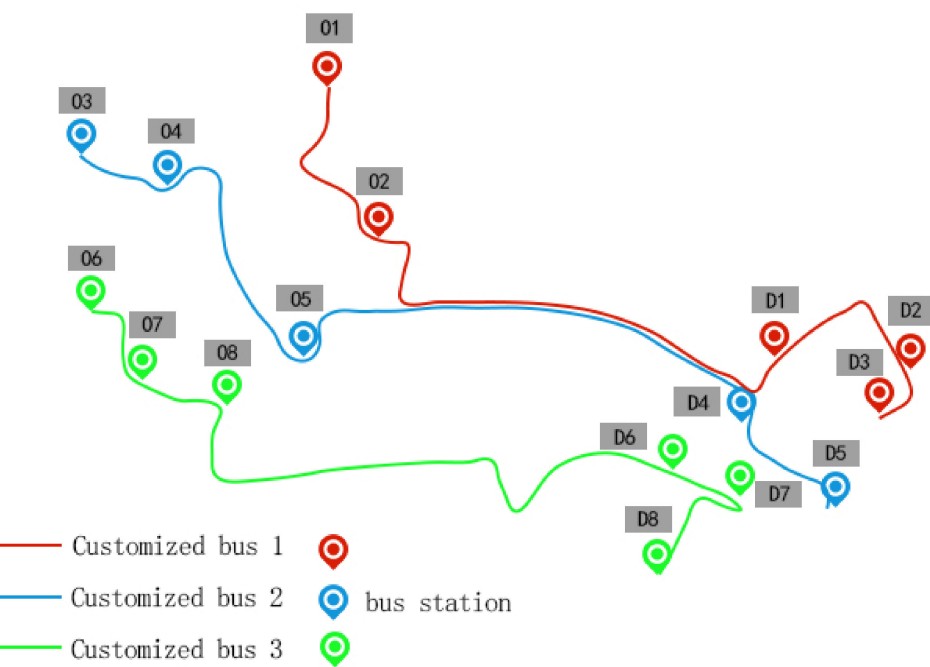

**Figure 2.** Schematic diagram of customized bus route.

For passengers taking customized buses, the comfort and convenience of taking customized buses have a greater impact on the competitiveness of customized buses in the market. The nonlinear coefficient of customized buses is also an important factor in determining whether the route is reasonable. The coefficient refers to the ratio of the length of the bus line of the linear distance between the first and last stations.

Among the three routes planned in this paper, line 1 has a total mileage of 6.4 km, a straight-line distance of 5.4 km from the first and last stop, and a nonlinear coefficient of 1.19; line 2 has a total mileage of 7.5 km, a straight-line distance between 7.1 km from the first and last stop, and a nonlinear coefficient of 1.06; and line 3 has a total mileage of 7.1 km, a straight-line distance of 5.3 km from the first and last stop, and a nonlinear coefficient of 1.34. These three lines are not too far from the station detour; to meet the comfort and direct transportation characteristics and service level of the customized bus, passengers choosing customized bus travel can have a better travel experience at a lower cost.

From the perspective of the bus company, the enterprise's fare revenue is 520 CNY, fixed cost is 300 CNY, and running cost is 45.65 CNY, which ensures a positive return to the enterprise. Applying the two-level model of customized bus route planning based on vehicle run time uncertainty established in this paper to custom bus route planning can take into account passengers' travel demands and quality of experience, while also bringing revenue to the enterprise, achieving a win–win situation.

## 6. Managerial Insights

The public travel service landscape is changing profoundly, and operating companies should continue to improve the experience and efficiency of customized bus travel, cutting costs and increasing efficiency to achieve high-quality development. This paper therefore considers factors such as differences in actual conditions due to the vehicle's travel speed road conditions during operation, the operating time of the vehicle between two stops, and the number of passengers getting on and off at each stop as uncertain variables and establishes a two-tier planning model with the maximum total revenue of the bus company as the upper-level objective and the minimum total passenger travel cost as the lower-level objective. The model ensures positive revenue for the company, increases the competitiveness of the bus under the impact of the metro, and enhances the confidence of

the company to implement customized bus services, while meeting the concept of green travel and sustainable development in the city.

## 7. Conclusions

This paper considers the uncertainties encountered in the operation of customized buses, analyzes the run time of buses between two stops as an uncertain variable, introduces uncertainty theory, establishes a two-level planning model with the maximum total revenue of the bus company as the upper-level objective and the minimum total travel cost of passengers as the lower-level objective, and demonstrates the feasibility of the model through cases. The established model takes into account various factors such as investment, revenue, route length, vehicle punctuality, and passenger capacity and is able to output a scientific custom bus routes to design to plan, which provides theoretical support for bus operators in bus route design. Customized public transportation in Nanchang is in the development stage, and linking the research ideas of this paper with the actual situation in Nanchang can provide new ideas of the development and future development of customized bus routes in Nanchang.

In real life, there are many different constraints or goals to be achieved in the optimization of customized bus routes, such as the bus configuration and route design of multiple models and the optimization of customized bus routes considering other uncertain factors. In addition, further optimization of the algorithm is also an important research direction in the future.

**Author Contributions:** All authors contributed equally to the article. All authors have read and agreed to the published version of the manuscript.

**Funding:** National Science Foundation of China (No. 52162042); National Science Foundation of China (No. 71961006); Project of Jiangxi Provincial Department of Education (No. GJJ190331).

**Conflicts of Interest:** The authors declare no conflict of interest.

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
