# Peer review of "Two-Level Planning of Customized Bus Routes Based on Uncertainty Theory"

_sustainability, doi:10.3390/su132011418_

Round 1

Reviewer 1 Report

See attached the MS word file including comments.

Author Response

请参阅附件

Reviewer 2 Report

In this paper, the authors have studied a customized bus route planning and dealt with various uncertainties associated with real life problem. The authors have formulated a mathematical model and maximizing the total profit revenue of bus company. It's a nice work. However, some of concerns regarding this paper to improve the quality and readability of paper as follows:

  1. The research questions should be described at the end of literature review section.
  2.  The authors have used GA to determine the objective function value (Max total profit revenue), but justification required why the authors have selected GA for this problem. There are so many other algo's, which performs better than GA.
  3. The operators description of GA are missing and How chromosomes are generated and what type of crossovers are used in this paper.
  4. There are so many typos and grammatical errors (i.e. equation no 18)
  5. Managerial insights should be included in section 6 and conclusion will move as section 7

On the basis of above points, I suggest it for minor revision.
